# Therapeutic Effects of Metaverse Rehabilitation for Cerebral Palsy: A Randomized Controlled Trial

**DOI:** 10.3390/ijerph20021578

**Published:** 2023-01-15

**Authors:** Ilyoung Moon, Yeongsang An, Seunghwa Min, Chanhee Park

**Affiliations:** 1Department of Physical Therapy, Yonsei University, Wonju 26493, Republic of Korea; 2Funrehab Co., Ltd., Daejeon 35229, Republic of Korea

**Keywords:** metaverse, virtual rehabilitation, cerebral palsy, gross motor function, cardiopulmonary function

## Abstract

Metaverse physical therapy (MPT), an adjuvant technology for the rehabilitation of children with cerebral palsy (CP), has gained notoriety in the clinical field owing to its accessibility and because it provides motivation for rehabilitation. The aim is to compare the gross motor function and cardiopulmonary function, the activities of daily living, quality of life (QOL), and the perceived risk of coronavirus disease (COVID)-19 transmission between MPT and conventional physical therapy (CPT). A convenience sample of 26 children with CP (mean age, 11.23 ± 3.24 years, 14 females) were randomized into either the MPT or CPT group and received therapy three days/week for four weeks. Clinical outcomes included gross-motor-function measure 66 (GMFM-66), heart rate (HR), Borg-rating perceived exertion (BRPE), functional independence measure (FIM), pediatric QOL, and the risk of COVID-19 transmission. An analysis of variance showed that compared with CPT, MPT exerted positive effects on GMFM, HR, and BRPE. An independent *t*-test showed that compared with CPT, MPT exerted positive effects on the perceived transmission risk of COVID-19 but not on FIM and QOL. Our results provide promising therapeutic evidence that MPT improves gross motor function, cardiopulmonary function, and the risk of COVID-19 in children with CP.

## 1. Introduction

Damage to the developing brain causes cerebral palsy (CP), an umbrella term for a diverse set of persistent but changeable movements and postural abnormalities [1,2,3]. In addition to movement difficulties, children with CP may experience challenges with communication, cardiopulmonary and gait function, and activities of daily living (ADL) [2,4,5]. The impact of CP on an individual is associated with lifespan, influencing independence in play, participation, and social and community activities [4,6].

In particular, spastic CP, which has hemiplegic, diplegic, and quadriplegic forms, is commonly linked to injury to the cortical motor region and white matter caused by hypoxia-ischemia [7]. Children with spastic CP have various balance and locomotor dysfunctions (asymmetric gait or stiff or scissor-like gait) depending on the severity of the sensorimotor impairment and the extremities involved, which predisposes them to falls and prevents them from participating in activities at school and in the community [4]. Cerebellar abnormalities, vermis damage, and genetic changes are linked to ataxic CP, which is the least common type of CP. Ataxic CP is described as a loss of coordination (as seen by issues with balance and walking) that results in considerable disability [1,2].

Currently, neurorehabilitation for CP involves direct, man-to-man, and hands-on therapy, with numerous studies reporting variable outcomes. Additionally, conventional physical therapy is associated with a variety of issues related to physical therapists and motivation [8]. Additionally, these issues have become more critical since the coronavirus disease (COVID)-19 pandemic [9]. There is an obvious need for an effective metaverse-based neurorehabilitation that involves minimal physical contact to prevent COVID-19 transmission during rehabilitation [10].

To mitigate the inherent limitations of current neurorehabilitation options for CP, we developed an innovative metaverse neurorehabilitation related to healthcare professionals. Metaverse rehabilitation can be rehabilitated anytime and anywhere, cost-free. However, conventional physical therapy varies greatly depending on labor intensiveness, physical stress, loss of motivation, cost, and differences in experience among physical therapists [11]. Metaverse rehabilitation comprises an artificial intelligence (AI)-based gross motor function classification system (GMFCS), movement evaluation through image recognition based on deep learning, the purchase of rehabilitation products as compensation through rehabilitation, and virtual character movement through weight shift. Metaverse neurorehabilitation is designed to provide corrective training with AI, decrease COVID-19 transmission risk, and increase fun and motivation [12,13].

Despite these potential physiotherapeutic advantages, conventional-physical-therapy-combined metaverse physical therapy (MPT) has not been investigated in CP rehabilitation. Therefore, the present study aimed to compare gross motor function and cardiopulmonary function, ADL, quality of life (QOL), and the perceived risk of COVID-19 transmission between MPT and conventional physical therapy (CPT). We hypothesized that MPT was as effective as CPT in gross-motor-function measure (GMFM), heart rate (HR), Borg-rating perceived exertion (BRPE), functional independence measure (FIM), pediatric QOL, and the risk of COVID-19 transmission.

## 2. Materials and Methods

### 2.1. Participants

A convenience sample of 26 children with CP was recruited from a community welfare center. The inclusion criteria were as follows: (1) the diagnosis of spastic and ataxic CP, (2) aged between 10 and 19 years, (3) children who can conduct classes with healthy children in elementary and middle schools, (4) GMFCS II–III for CP, (5) those who can directly respond to mediation and surveys, and (6) the ability to follow instructions. The exclusion criteria were as follows: (1) having epilepsy or taking antiepileptic drugs, (2) a history of trauma or surgery within the last six months, (3) severe cognitive or visual impairments, and (4) cardiopulmonary system impairments.

### 2.2. Ethical Assumptions

This study was approved by the Dajeon community center institutional review board (No. IRB-2022-06). Participants were selected in accordance with the ethical standards of the Committee on Human Experimentation of the institution in which the experiments were carried out or in accordance with the Declaration of Helsinki. Written informed consent was obtained from the parents or guardians of the children. After the participants were recruited via bulletin-board notices within the community center and presentation to parents and children, initial screening was conducted to determine whether the potential patients met the inclusion criteria. Informed consent was obtained from all the patients before participation.

### 2.3. Experimental Procedure

The present study used an experimental design in which all of the participants completed the pre- and post test. Clinical outcome tests included standardized GMFM, HR, BRPE, FIM, QOL, and a post-questionnaire to assess the risk of COVID-19 transmission. These tests were consistently implemented at the pretest and posttest.

#### 2.3.1. GMFM 66

The GMFM was designed to measure changes in gross motor function in children with CP; it assesses motor function (how much of the task the child can perform) rather than the quality of motor performance (how well the child performed the task). The GMFM comprises five major testing sub-items: (A) lying and rolling; (B) sitting; (C) crawling and kneeling; (D) standing; and (E) walking, running, and jumping. The scoring system ranges from 0 to 3, where 0 indicates ‘no initiation’, 1 indicates ‘initiation’, 2 indicates ‘partial completion’, and 3 indicates ‘completion’ [14]. The intra- and inter-tester reliability for GMFM are well established, with intraclass correlation coefficients (ICC) of 0.96 and 0.97, respectively [15].

#### 2.3.2. Cardiopulmonary Function

A 40-m track with a firm, level, and straight surface was used for the 6-min walk test. Within the specified 6-min period, each patient was permitted to pause and take a break as needed. The walking distance was calculated after the completion of the task. A portable monitor was used to continually record the patient’s HR during the 6-min walk test. Following this test, patients reported their felt degree of exertion on a number scale ranging from 6 (no exertion) to 20 (maximal exertion) [16]. BRPE is used to subjectively quantify an individual’s perception of the physical demands of an activity. The evaluation of cardiopulmonary fitness endurance using BRPE is an indirect approach. The validity of BRPE is well established, with r = 0.80–0.90 [17].

#### 2.3.3. FIM

The FIM is an outcome measure of the severity of disability for inpatient rehabilitation. It rates 18 ADLs on a 7-point scale ranging from 1 (“fully dependent”) to 7 (“independent with no aids”). The FIM comprises 18 measures, with subscales measuring social cognition (3 items), transfers (3 items), locomotion (2 items), sphincter control (2 items), self-care (6 items), and communication (2 items) [18]. The validity and reliability of the FIM are well established with a Kappa coefficient = 0.92 and an ICC = 0.99, respectively [19].

#### 2.3.4. Pediatric QOL

QOL is a brief measure of the health-related QOL of children with CP. There are 23 items total on the questionnaire, including eight for physical functioning, five for emotional functioning, five for social functioning, and five for school functioning. Each question asks for a response on a 5-point scale: 0 (“never a problem”), 1 (“nearly never a problem”), 2 (“often a problem”), 3 “frequently a problem”), and 4 (“almost always a problem”) [20]. The validity and reliability of the QOL are well established, with Cronbach’s α = 0.93 and ICC = 0.89, respectively [21,22].

#### 2.3.5. Perceived Risk of COVID-19 Transmission

The perceived transmission risk of COVID-19 section had questions on social distancing and COVID-19 transmission. The scale ranged from 0 (“COVID-19 free”) to 10 (“maximal COVD-19”). All participants in both interventions participated in the survey [23].

### 2.4. Intervention

Participants were randomly assigned to either the MPT or CPT group, and both groups received 30 additional minutes at each session, 3 days/week, for 4 weeks. Coin flipping was used to assign patients to either the control or experimental group.

The CPT group received the standard neurodevelopmental treatment (NDT) framework and clinical evidence. With the advancement of overground gait training with or without assistance aids and motor control, CPT comprises mobility and stability exercises based on muscle stretching and strengthening and dynamic or static balance [8]. Additional time-standard physical therapy focuses on ambulation and gross-motor-function training activities. Regarding intervention strategies, licensed physical therapists provided interventions based on the evaluation of selected clinical evidence [8,24].

The MPT group received the standard NDT, including a 30-min CPT session and an additional 30-min MPT. Specifically, the MPT concept involves children-specific training according to GMFCS level standards. Parents downloaded the metaverse application to their smartphones or tablets. Each parent had a smart device and an internet connection of at least one megabyte per second to download the program. To eliminate measurement errors, the distance between the subject and the smartphone was fixed at 1 m [25]. The parents objectively investigated the child’s function according to the GMFCS level-based design, and the physical therapist asked questions again to ensure accuracy. A questionnaire was used to gather some information for the MPT implementation. The MPT questionnaire items were as follows: the child’s name, date of birth, type of CP, presence or absence of assistance aid, risk of falls, cognitive level, orthosis wear, spasticity, and GMFCS level. Specifically, the MPT concept was based on home program instruction sheets for infants and children according to an AI-based diagnosis (Figure 1) [26]. Based on the survey results, AI provided the most effective individualized exercise. MPT was provided using an individualized intervention video, which outlines the individualized exercise as outlined in the evidence along with the video. After watching the video, parents evaluated whether the child followed well and whether the exercise seemed suitable for the child. If the exercise was difficult to follow, AI adjusted and recommended other exercises (Figure 2). The deep learning convolutional neural network (CNN) algorithm was built to update the intervention program based on information obtained from the reassessment and weighing of the exercise in accordance with the audiovisual feedback [27]. After watching the video, the CNN-based image recognition system provided audiovisual feedback on the quality of the movement and any problematic areas so that the participant could perform a customized exercise by repeating the right action [28]. AI-based metaverse neurorehabilitation determines the diagnosis and exercise program based on subjective information. The deep learning algorithm used in this study was based on a CNN [29]. It is the most commonly used AI algorithm in the clinical field and is capable of learning from historical examples, analyzing nonlinear data, and handling imprecise information. After performing the exercise, the program is given coins that can be used by the participants within the metaverse. When a child performs a walking motion, the base camera installed with the program recognizes it and can go to a pharmacy store in the metaverse, such as gymball and TheraBand, to purchase the necessary items for rehabilitation. In addition, if a child walks while looking at the camera installed with the program, the motion is recognized, and the virtual character can enter a hospital in the metaverse. Children can go anywhere in the metaverse and meet friends. If children access each other at the same time, they can meet each other separately in the metaverse rehabilitation space. Without the need for VR glasses or something, the smartphone or tablet camera was used to recognize motion and access virtual reality.

### 2.5. Statistical Analysis

Descriptive statistics included mean and standard deviation (SD). We used G-power software (version 3.1) to assess the sample size based on our previous study [30]; the G-power software generated 24 participants with CP, calculated from the medium effect size and power (1 − β = 0.8). All continuous variables were analyzed, using the Kolmogorov–Smirnov test, assuming a normal distribution. Analysis of variance (ANOVA) was used to assess outcome measurement variables, including GMFM, HR, BRPE, FIM, and QOL. An independent *t*-test was used to assess the perceived risk of COVID-19 transmission. If there was a significant time or difference, a post hoc t-test was performed to compare changes in variables from the pre- and post test within and between group differences. If an interaction effect was observed, a post hoc *t*-test was implemented. All statistical analyses were performed using SPSS version 26 (SPSS Inc., Chicago, IL, USA). The statistical significance level was set at *p* < 0.05.

## 3. Results

### 3.1. Demographic and Clinical Characteristics of Patients (N = 26)

The demographic and clinical characteristics of patients are shown in Table 1. No significant differences were observed between the CPT and MPT groups in this regard.

### 3.2. Clinical Outcome Measurements

#### 3.2.1. GMFM

Repeated-measures ANOVA showed a significant time effect (*p* = 0.001) for the GMFM. The post hoc test revealed significant differences in GMFM scores at pre- and post test in both groups (*p* = 0.01) (Table 2).

#### 3.2.2. Cardiopulmonary Function

Repeated-measures ANOVA revealed a significant time × group interaction (*p* = 0.001), time effect (*p* = 0.03), and between-group difference (*p* = 0.01) for HR. The interaction post hoc test revealed that MPT showed a greater increase in HR than CPT did. A post hoc *t*-test revealed significant differences in HR scores between the MPT pre- and post test (Table 3).

Repeated-measures ANOVA indicated significant effects of both CPT and MPT on the BPRE score (*p* = 0.001) and a significant difference in the MBI score between the two groups (*p* = 0.03). A paired *t*-test revealed significant differences in BPRE scores between the MPT pretest and posttest (*p* = 0.001). Post hoc analysis showed that MPT caused a greater increase in BRPE than in CPT (*p* = 0.001) (Table 3).

#### 3.2.3. FIM and QOL

Repeated-measures ANOVA did not show a significant time effect or a significant between-group or time × group interaction for FIM (*p* > 0.05) in either group (Table 4).

Repeated-measures ANOVA did not show a significant time effect or a significant between-group or time × group interaction for QOL (*p* > 0.05) in either group (Table 4).

### 3.3. Perceived Risk of COVID-19 Transmission

The independent *t*-test showed a significantly greater decrease in perceived COVID-19 contamination with MPT than with CPT (*p* = 0.001), indicating a greater decrease in COVID-19 contamination with MPT than with CPT during the COVID-19 pandemic (Figure 3).

## 4. Discussion

To our knowledge, this is the first study on physical therapeutic intervention using metaverse-based self-management rehabilitation that is aimed at assessing and comparing the effects of CPT and MPT on gross motor function, cardiopulmonary function, ADL, QOL, and the transmission risk of COVID-19 in patients with CP. As hypothesized, conventional-physical-therapy-combined MPT was as effective as CPT in terms of GMFM, HR, BRPE, FIM, QOL, and perceived risk of COVID-19 transmission. Most importantly, conventional-physical-therapy-combined MPT was better compared with CPT in terms of reducing the termination of COVID-19 and improving cardiopulmonary function.

Gross motor analysis showed that improvement in the GMFM was significantly different between the MPT and CPT. This finding was consistent with the results of Arnoni et al. (2019), who reported a 1.45–23.32% increase in gait parameters and GMFM D and E scores after virtual reality (VR) rehabilitation compared with CPT in 15 children with CP [31]. Moreover, a 7% improvement in GMFM D and E scores following VR intervention compared with pre-intervention was found in four adolescents with CP. A 1.58% increase in GMFM-88 scores after horse-riding with VR has also been reported in 17 children with CP [32].

Improvements in gross motor function indicate the feasibility of task-oriented training using VR training strategies for gross motor skills in patients with CP. Our results suggest that a controlled virtual metaverse can simulate the components of the real (physical) world and provide functional motor gains. Indeed, following motor training in a virtual environment, patients’ functional movements could be adjusted based on environmental information by the exploration and selection of the most appropriate motor action [33]. Together, the use of audiovisual feedback (provided through contact with the virtual environment) and constant therapist-provided hints during metaverse therapy are possibly the main factors guiding adaptive responses to the virtual environment. The need to accomplish tasks in a virtual environment (by an active video game with body cognition) challenges the sensory and neuromotor systems. The visual system is responsible for constantly adjusting the attention required to accomplish activities [33].

Cardiopulmonary function analysis revealed a significant improvement in HR and BRPE after metaverse rehabilitation compared to CPT. This finding was consistent with that by Chuang et al. (2006), who reported that the VR group had a greater increase in HR and metabolic target than the non-VR group among 20 patients with post-coronary artery bypass graft [34]. Cacau et al. (2013) found that the VR-augmented group experienced decreased pain, with better functional capacity and 6-min walking distance compared with the conventional post-operation physical therapy group in 60 patients’ post-cardiac surgery [35]. The results indicated that using a VR device or virtual environment during stationary cycling can reduce sympathetic tone and thus increase blood flow to the muscles, prolonging exercise duration and enhancing cardiopulmonary fatigue resistance [36]. Moreover, metaverse technology has been shown to affect psychological responses. VR technology significantly decreases patient anxiety and tension, both of which are counterproductive to the outcomes of cardiopulmonary rehabilitation [37,38].

Analysis of the perceived COVID-19 transmission risk indicated that improvement in the perceived COVID-19 transmission risk was significantly different after metaverse rehabilitation compared with CPT. These findings suggest that the smartphone application approach is more suitable for reducing the risk of potential COVID-19 transmission during a pandemic. Through remote rehabilitation via a metaverse, patients can learn to control their functional movements by modifying their behavior.

Metaverse intervention with a user-friendly interface ensures that children in all sociocultural environments can be easily incorporated into the intervention program. Metaverse rehabilitation could make exercise more enjoyable, mainly because of the virtual environment, deep learning-based exercise, and audiovisual feedback.

This study has several limitations, which should be considered in future research. First, the lack of follow-up evaluation might have had important effects on the sustainable physiotherapeutic effects of MPT in children with CP. Second, we did not compare the GMFCS (which influences the ability of children with CP to carry out ADLs) with gait ability. Third, there may be differences in movement recognition depending on the quality of the smartphone or tablet camera.

## 5. Conclusions

This therapeutic study has compared the effects of MPT and CPT in patients with CP. We demonstrated that MPT was effective in improving gross motor function and cardiopulmonary function and perceived COVID-19 transmission risk. Clinically, our results provide encouraging evidence that MPT is more effective than CPT in the management of children with CP. Our findings provide important insights that will guide therapists when designing an effective pediatric physical therapy model for children with CP. Even children who have difficulty accessing hospitals can have the same effect as CPT. In addition, even when it is difficult to visit a hospital due to an infectious disease, it is as effective as conventional physical therapy.

## Figures and Tables

**Figure 1 ijerph-20-01578-f001:**
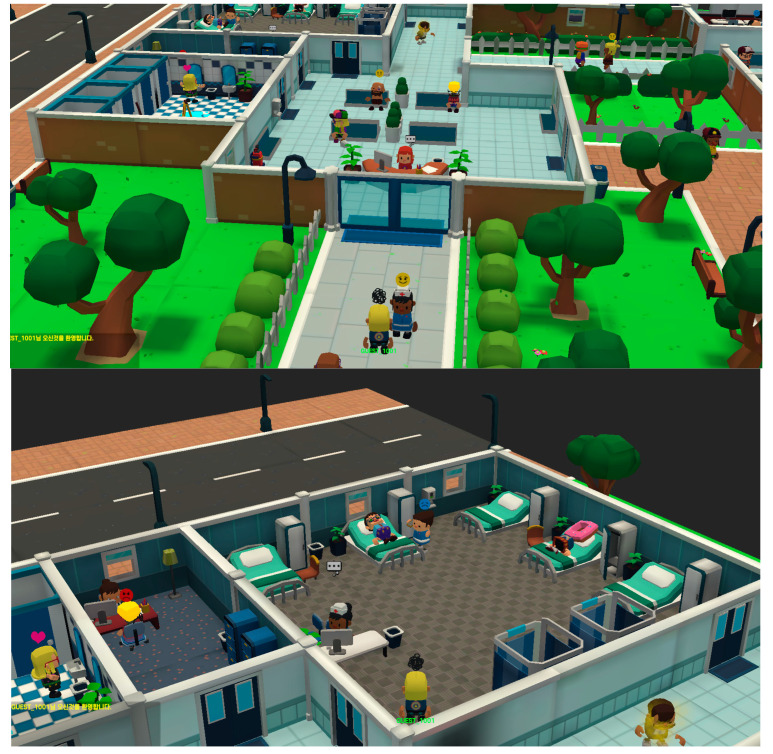
Metaverse rehabilitation.

**Figure 2 ijerph-20-01578-f002:**
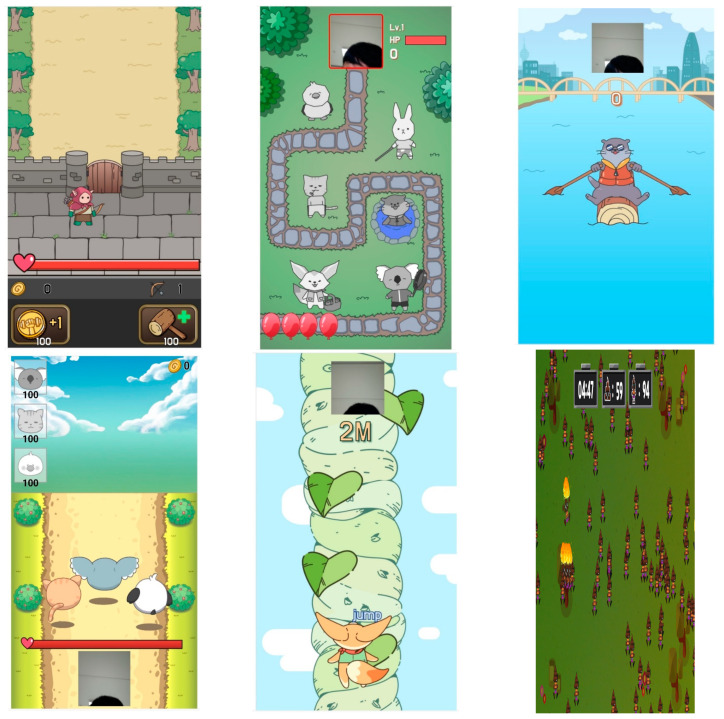
Metaverse rehabilitation game.

**Figure 3 ijerph-20-01578-f003:**
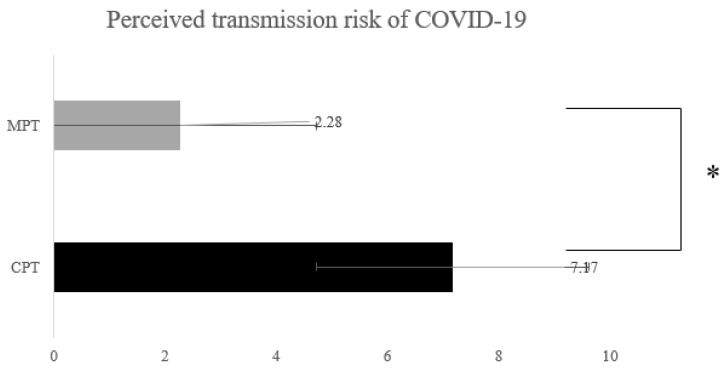
Perceived transmission risk of COVID-19. CPT: Conventional physical therapy. MPT: conventional-physical-therapy-combined metaverse physical therapy. COVD19; Corona virus disease 2019. * denotes *p* < 0.05.

**Table 1 ijerph-20-01578-t001:** Demographic and clinical characteristics of patients.

	CPT ^a^ Group (n = 13)	MPT ^b^ Group (n = 13)	*p-*Value
Sex (male/female)	6/7	6/7	1.00
Age (years)	16.15 ± 3.16	17.43 ± 2.88	0.36
Body height (cm)	133.17 ± 21.41	140.21 ± 18.11	0.14
Body mass (kg)	38.44 ± 16.08	42.56 ± 20.37	0.10
^c^ CP classification			
Spastic/ataxic	10/3	11/2	0.64

^a^ CPT: Conventional physical therapy. ^b^ MPT: Conventional-physical-therapy-combined metaverse physical therapy. ^c^ CP: Cerebral palsy.

**Table 2 ijerph-20-01578-t002:** Gross-motor-function measure.

	CPT ^b^		MPT ^c^		*p*-Value		
	Pre Test	Post Test	Pre Test	Post Test	Time Effect	Between Groups	Time × Group Interaction
GMFM ^a^	68.13 ± 18.27	72.66 ± 20.17	70.16 ± 11.37	75.01 ± 18.16	0.001 *	0.21	0.48

^a^ GMFM: Gross-motor-function measure. ^b^ CPT: Conventional physical therapy. ^c^ MPT: Conventional-physical-therapy-combined metaverse physical therapy. * denotes *p* < 0.05.

**Table 3 ijerph-20-01578-t003:** Cardiopulmonary function.

	CPT ^c^		MPT ^d^		*p*-Value		
	Pre Test	Post Test	Pre Test	Post Test	Time Effect	Between Groups	Time × Group Interaction
HR ^a^	78.16 ± 4.88	79.14 ± 3.17	74.84 ± 5.12	81.44 ± 4.11	0.03 *	0.17	0.001 *
BRPE ^b^	11.14 ±0.88	10.88 ±1.43	12.26 ±1.10	10.13 ±0.75	0.001 *	0.03 *	0.02 *

^a^ HR: Heart rate. ^b^ BRPE: Borg-rating perceived exertion scale. ^c^ CPT: Conventional physical therapy. ^d^ MPT: Conventional-physical-therapy-combined metaverse physical therapy. * denotes *p* < 0.05.

**Table 4 ijerph-20-01578-t004:** Functional independence measure and quality of life.

	CPT ^c^		MPT ^d^		*p*-Value		
	Pre Test	Post Test	Pre Test	Post Test	Time Effect	Between Groups	Time × Group Interaction
FIM ^a^	17.88 ± 8.33	18.21 ± 6.42	16.79 ± 9.77	18.33 ± 11.21	0.17	0.33	0.18
QOL ^b^	56.21 ± 17.44	53.19 ± 14.66	54.88 ± 16.71	52.67 ± 15.85	0.09	0.51	0.30

^a^ FIM: Functional independence measure. ^b^ QOL: Pediatric quality of life. ^c^ CPT: Conventional physical therapy. ^d^ MPT: Conventional-physical-therapy-combined metaverse physical therapy.

## Data Availability

Data supporting reported results can be requested from the corresponding author (chaneesm@yonsei.ac.kr).

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
