# Peer review of "Therapeutic Effects of Metaverse Rehabilitation for Cerebral Palsy: A Randomized Controlled Trial"

_ijerph, 2023, doi:10.3390/ijerph20021578_

Round 1
Reviewer 1 Report
Dear Authors,
Congratulations on your manuscript. Telerehabilitation has long been a new healthcare trend, with promising functional outcomes, especially considering homecare. Using VR and AR was another big step. With metaverse, new options arise, and thus your research provides interesting insights, namely when you are dealing with a very vulnerable population, as children with CP are.
I have some comments and suggestions to help you improve your work.
TITLE
- According to relevant guidelines (see EQUATOR Network), the title should contain a common expression for the study type. When you write «A State of the Art» the reader will think this is a literature review. Please, change the title according to this suggestion.
ABSTRACT
- It's hard to follow, namely when you write «To compare the gross (...)», but don't introduce it as your aim. Write something like: «The aim is to compare (...)».
CITING STYLE: check citing style according to IJERPH's guidelines.
INTRODUCTION
Your Introduction can be improved. Please, see the following suggestions:
- In the third paragraph, line 44, you mention that numerous studies have reported variable outcomes. Would be interesting to briefly mention which ones, and relate them to your study.
- In that same paragraph, between lines 44 and 50, you present a justification for the need for MTP and, therefore, for your research, as being problems related to healthcare professionals, instead of problems related to CP or ADL in children. You should focus on your population - children with CP, ADL performance, etc. - instead of describing a new problem, for which you won't give a solution in your article.
- In line 52, you mention that current neurorehabilitation options for CP have some inherent limitations, but you don't describe them. Would be very useful, to understand the possible advantage of MTP.
- In the last paragraph of the Introduction (lines 52 to 60), you should explore in more detail on why CPT is insufficient for children with CP, thus giving the motivation for the development of MPT.
MATERIALS AND METHODS
- Mean age and sex are a result. Please move to the adequate section.
- I suggest creating a specific subsection for the ethical approvals (between lines 71 and 74). Something like: «Ethical Assumptions».
- When you mention the institutional review board, you should at least state which one: the community welfare center or another one? You can write its name also.
- More can be detailed about how recruitment was performed: presentation to parents, any specific adaptation to allow some sort of presentation to the children themselves, etc.
- Please, describe the first time they appear in the text: GMFM, HR, BRPE, FIM, QOL.
- Regarding written consent, if children can answer self-reported questionnaires (did I understand wrong that they did do answer themselves some questionnaires, or were solely the parents?), why wouldn't children with more than 16 years also sign their own consent? It really depends on the severity of CP and cognitive status. Could you say more about this?
- Still related to the previous comment, in 2.2.5 you're reporting a perception. I was wondering if children with CP, as young as 10 years, depending on the severity, can really report something so subjective. Could this be an important limitation of your study?
- Regarding 2.3, how was randomization performed? Any software? Also, from line 146 onwards, when you say that the MPT group also received a 30-min CPT session, can we conclude that MPT always needs some sort of CPT session to be efficient and achieve the desired outcomes? Could you describe this earlier in the paper?
- In line 154 you mention a questionnaire: «The questionnaire items were as follows (...)». Which questionnaire is this?
- You mention: «AI provided the most effective individualized exercise» and then Figure 1. But the figure is somehow poor and doesn't allow the reader to speculate about effective exercises. What does Figure 1 really describe? Couldn't you add some explanatory letters or words to it? Also, when you say there's an AI diagnosis and an individualized exercise, is that because the image you present is different for each child?
- How was the MPT software app developed? Was it previously tested? You do mention a pilot test, but with no further details, like if you had to change anything to comply with the piloting results.
- At the end of 2.3, you briefly describe some kind of interaction of the children in the metaverse (e.g., going to gym ball). Does this mean that children can interact with other children? It's not clear, mainly because of your images, if the metaverse is accessed through VR glasses or something. Or just a mobile app? Could you describe this in more detail?
- Title of Figure 2 is duplicated.
DISCUSSION
- In the first paragraph, line 264, I think you want to write CPT instead of MPT: «As hypothesized, MPT was as effective as CPT (...)».
CONCLUSIONS
- Try to rewrite it to clearly answer your main objectives and hypothesis.
- It would be more useful for future research if you clearly describe with important insights are you mentioning here.
REFERENCES
- 65% of your references (28 of 43) have 10 years or more (I'm also considering 2013 and 2014 outdated). This isn't reasonable for this kind of topic. Could you update some of them? Have a maximum of 5% of references with 10 years or more.
Wishing you a Happy Xmas and New Year,
Author Response
Reviewers comments 1
Congratulations on your manuscript. Telerehabilitation has long been a new healthcare trend, with promising functional outcomes, especially considering homecare. Using VR and AR was another big step. With metaverse, new options arise, and thus your research provides interesting insights, namely when you are dealing with a very vulnerable population, as children with CP are.
I have some comments and suggestions to help you improve your work.
TITLE
- According to relevant guidelines (see EQUATOR Network), the title should contain a common expression for the study type. When you write «A State of the Art» the reader will think this is a literature review. Please, change the title according to this suggestion.
Authors response: This was revised (Title).
ABSTRACT
- It's hard to follow, namely when you write «To compare the gross (...)», but don't introduce it as your aim. Write something like: «The aim is to compare (...)».
Authors response: This was revised (Line 10).
CITING STYLE: check citing style according to IJERPH's guidelines.
Authors response: This was revised.
INTRODUCTION
Your Introduction can be improved. Please, see the following suggestions:
- In the third paragraph, line 44, you mention that numerous studies have reported variable outcomes. Would be interesting to briefly mention which ones, and relate them to your study.
Authors response: This was revised (Lines 42-44).
- In that same paragraph, between lines 44 and 50, you present a justification for the need for MTP and, therefore, for your research, as being problems related to healthcare professionals, instead of problems related to CP or ADL in children. You should focus on your population - children with CP, ADL performance, etc. - instead of describing a new problem, for which you won't give a solution in your article.
Authors response: This was revised (Lines 49-50).
- In line 52, you mention that current neurorehabilitation options for CP have some inherent limitations, but you don't describe them. Would be very useful, to understand the possible advantage of MTP.
Authors response: This was revised (Lines 50-52).
- In the last paragraph of the Introduction (lines 52 to 60), you should explore in more detail on why CPT is insufficient for children with CP, thus giving the motivation for the development of MPT.
Authors response: This was revised (Lines 50-52).
MATERIALS AND METHODS
- Mean age and sex are a result. Please move to the adequate section.
Authors response: This was revised (Line 69).
- I suggest creating a specific subsection for the ethical approvals (between lines 71 and 74). Something like: «Ethical Assumptions».
Authors response: This was revised (Lines 77-87).
- When you mention the institutional review board, you should at least state which one: the community welfare center or another one? You can write its name also.
Authors response: This was revised (Lines 77-87).
- More can be detailed about how recruitment was performed: presentation to parents, any specific adaptation to allow some sort of presentation to the children themselves, etc.
Authors response: This was revised (Lines 77-87).
- Regarding written consent, if children can answer self-reported questionnaires (did I understand wrong that they did do answer themselves some questionnaires, or were solely the parents?), why wouldn't children with more than 16 years also sign their own consent? It really depends on the severity of CP and cognitive status. Could you say more about this?
Authors response: Subjects were selected so that participants could do it themselves (Lines 71-73).
- Still related to the previous comment, in 2.2.5 you're reporting a perception. I was wondering if children with CP, as young as 10 years, depending on the severity, can really report something so subjective. Could this be an important limitation of your study?
Authors response: Subjects were selected so that participants could do it themselves (Lines 71-73).
- Regarding 2.3, how was randomization performed? Any software? Also, from line 146 onwards, when you say that the MPT group also received a 30-min CPT session, can we conclude that MPT always needs some sort of CPT session to be efficient and achieve the desired outcomes? Could you describe this earlier in the paper?
Authors response: This was added (Lines 139-140). This was revised (Lines 59-60; 267-271).
- In line 154 you mention a questionnaire: «The questionnaire items were as follows (...)». Which questionnaire is this?
Authors response: This was revised (Lines 155-158).
- You mention: «AI provided the most effective individualized exercise» and then Figure 1. But the figure is somehow poor and doesn't allow the reader to speculate about effective exercises. What does Figure 1 really describe? Couldn't you add some explanatory letters or words to it? Also, when you say there's an AI diagnosis and an individualized exercise, is that because the image you present is different for each child?
Authors response: Explanatory letters were deleted because of the high plagiarism rate, and customized exercises tailored to the child's function were prescribed. Each child presents a different image.
- How was the MPT software app developed? Was it previously tested? You do mention a pilot test, but with no further details, like if you had to change anything to comply with the piloting results.
Authors response: We previously developed a similar program to approach adolescents and extracted sample sizes based on the data.
- At the end of 2.3, you briefly describe some kind of interaction of the children in the metaverse (e.g., going to gym ball). Does this mean that children can interact with other children? It's not clear, mainly because of your images, if the metaverse is accessed through VR glasses or something. Or just a mobile app? Could you describe this in more detail?
Authors response: This was revised (Lines 182-185).
- Title of Figure 2 is duplicated.
Authors response: This was revised.
DISCUSSION
- In the first paragraph, line 264, I think you want to write CPT instead of MPT: «As hypothesized, MPT was as effective as CPT (...)».
Authors response: This was revised (Lines 267-271).
CONCLUSIONS
- Try to rewrite it to clearly answer your main objectives and hypothesis.
Authors response: This was revised (Lines 267-271).
- It would be more useful for future research if you clearly describe with important insights are you mentioning here.
Authors response: This was revised (Lines 328-331).
REFERENCES
- 65% of your references (28 of 43) have 10 years or more (I'm also considering 2013 and 2014 outdated). This isn't reasonable for this kind of topic. Could you update some of them? Have a maximum of 5% of references with 10 years or more.
Authors response: This was revised (References)

Reviewer 2 Report
This paper describes the use of distant type of the rehabilitation program for CP children, based on the games. In this program the difficulty level (type of the video game) was assessed by the AI based on the questionnaire filled by parents and physiotherapist before the program. This type of the rehabilitation intervention can be valuable not only in times of the pandemic, but also for children living in places distant from the health care facilities, where the access to the rehabilitation is limited. Therefore this paper can be interesting for the readers of the journal. Before its publication some issues should be addressed.
1. There is no information about who filled the QOL questionnaires: parents, children or physiotherapists. The CP children recruited to the study were from 10 to 19 years old, thus if their mental development is not affected they could fill the questionnaires. The perception of the parents and patients of the quality of life can be different, therefore such information is important. The information about the patients’ mental development is also important, as the understanding and following instruction could affect their ability to perform the intended in the game roles.
2. The parents installed the rehabilitation program either on the phones or tablets. The devices’ cameras were used. The information from these cameras were used to control the avatars in the game (line 175 – 180). The quality of the cameras were probably different therefore the patients’ movement recognition could also be different. Some comments / discussion about this factor should be given in the Discussion.
3. The authors used only parametric statistics (means, standard deviation, analysis of variance, t-test). Was the normality of the variables checked?
Author Response
This paper describes the use of distant type of the rehabilitation program for CP children, based on the games. In this program the difficulty level (type of the video game) was assessed by the AI based on the questionnaire filled by parents and physiotherapist before the program. This type of the rehabilitation intervention can be valuable not only in times of the pandemic, but also for children living in places distant from the health care facilities, where the access to the rehabilitation is limited. Therefore this paper can be interesting for the readers of the journal. Before its publication some issues should be addressed.
- There is no information about who filled the QOL questionnaires: parents, children or physiotherapists. The CP children recruited to the study were from 10 to 19 years old, thus if their mental development is not affected they could fill the questionnaires. The perception of the parents and patients of the quality of life can be different, therefore such information is important. The information about the patients’ mental development is also important, as the understanding and following instruction could affect their ability to perform the intended in the game roles.
Authors response: All surveys and intervention were conducted directly by the participants, and only those who met the inclusion criteria were recruited (Lines 71-73).
- The parents installed the rehabilitation program either on the phones or tablets. The devices’ cameras were used. The information from these cameras were used to control the avatars in the game (line 175 – 180). The quality of the cameras were probably different therefore the patients’ movement recognition could also be different. Some comments / discussion about this factor should be given in the Discussion.
Authors response: This was revised (Lines 319-320).
- The authors used only parametric statistics (means, standard deviation, analysis of variance, t-test). Was the normality of the variables checked?
Authors response: This was revised (Lines 202-203).

Round 2
Reviewer 1 Report
Dear authors,
Congratulations on the review and work performed.
I'm endorsing the publication of this manuscript.
Best regards,
Reviewer 2 Report
The authors addressed the raised in the previous review concerns, and introduced new information. In my opinion the papaer is now ready for the publication.